# Graphene-Based Membranes for Water Desalination: A Literature Review and Content Analysis

**DOI:** 10.3390/polym14194246

**Published:** 2022-10-10

**Authors:** Yexin Dai, Miao Liu, Jingyu Li, Ning Kang, Afaque Ahmed, Yanping Zong, Jianbo Tu, Yanzhen Chen, Pingping Zhang, Xianhua Liu

**Affiliations:** 1School of Environmental Science and Engineering, Tianjin University, Tianjin 300354, China; 2Tianjin Marine Environmental Center Station, Ministry of Natural Resources, Tianjin 300450, China; 3College of Food Science and Engineering, Tianjin Agricultural University, Tianjin 300384, China

**Keywords:** graphene, membrane, desalination, bibliometric analysis, reverse osmosis, forward osmosis, solar water purification

## Abstract

Graphene-based membranes have unique nanochannels and can offer advantageous properties for the water desalination process. Although tremendous efforts have been devoted to heightening membrane performance and broadening their application, there is still lack of a systematic literature review on the development and future directions of graphene-based membranes for desalination. In this mini-review, literature published between 2011 and 2022 were analyzed by using the bibliometric method. We found that the major contributors to these publications and the highest citations were from China and the USA. Nearly 80% of author keywords in this analysis were used less than twice, showing the broad interest and great dispersion in this field. The recent advances, remaining gaps, and strategies for future research, were discussed. The development of new multifunctional nanocomposite materials, heat-driven/solar-driven seawater desalination, and large-scale industrial applications, will be important research directions in the future. This literature analysis summarized the recent development of the graphene-based membranes for desalination application, and will be useful for researchers in gaining new insights into this field.

## 1. Introduction

The shortage and security of water resources are the main challenges in the 21st century. The rapid growth of population, the fast development of industrialization, and accelerated climate change has caused great pressure on the security of water resources, and further exacerbated the global water crisis [1,2,3,4,5,6]. The world’s water council predicts that the number of people affected by water shortage will rise to 3.9 billion in the coming decades. As one of the most promising solutions to mitigate the water crisis, desalination will expand water availability beyond that of the hydrological cycle. Seawater desalination converts seawater into a usable water product for industries or even domestic applications, providing an unlimited, continuous source of high-quality water without affecting existing freshwater habitats. Commercial desalination of water started during the early part of the 20th century. With the growing demand for water and the lack of sustainable natural water resources, reliance on desalination will continue to rise and, as a result, its energy consumption and environmental effects will continuously increase without limitations. Overall, over 75 million people worldwide are estimated to be obtaining fresh water by desalinating seawater or brackish water. Many desalination technologies, such as distillation [7,8,9,10,11], reverse osmosis (RO) [12,13,14,15,16], and electrodialysis [17,18,19], have been developed. The general application of polyamide membrane-based RO desalination seems to be well established, but it is worth noting that the most ideal membrane should have the highest possible water flux, complete salt rejection, high anti-fouling, and oxidation tolerance. Therefore, it is still an urgent task for researchers around the world to explore more advanced seawater desalination materials and processes to further improve the performance of seawater desalination to the maximum extent. On the one hand, RO membranes based on polyamide membranes have been extensively studied and modified to maximize their desalination performance. Various surface modifications, including grafting, deposition, and merging, have been widely reported [20,21]. Alternatively, new desalination membranes have emerged, some of which have shown great potential. For example, carbon materials [22], molybdenum disulfide (MoS_2_) [23], and MXene membranes [24], show great potential in seawater desalination.

With its unique two-dimensional structure and fast mass transfer channels, graphene materials have received extensive attention for their applications in membrane separation, such as water purification, solvent dehydration, and gas separation [25,26]. Despite some limitations, graphene-based membranes have great advantages compared to commercial polyamide membranes, such as nanoporous graphene membranes with 2–3 orders of magnitude higher water flux than conventional RO membranes and salt rejections close to 100% [27]. In addition, the high surface hydrophilicity of GO membranes is beneficial to mitigate membrane fouling, especially biological fouling [28]. Energy consumption and costs will be significantly reduced by developing anti-fouling membranes with surface properties that inhibit the adhesion of impurities. Apart from this, graphene and derived materials possess other interesting properties, such as high thermal conductivity of about 5000 W m^−1^K^−1^, making them ideal candidates for solar-driven photothermal desalination processes [29]. They have good electrical conductivity, which means they can be used as electrodes for capacitive deionization of seawater [30]. Until now, there were several studies with different emphasis summarizing graphene-based membranes for seawater desalination [17,31,32,33,34,35], and no insights were given into the key criteria of the graphene-based for the desalination process. In addition, there is still a lack of a systematic literature review by evaluating the field via a bibliometric approach to group existing studies and develop new insights.

With the rapid development of scientometrics, and informatics technology, the quantification and visualization of literature review research have rapidly developed. The knowledge graph was a literature analysis method that has emerged in recent years [36,37]. This method has the advantages of processing a large number of documents, intuitive visualization, and diverse analysis angles, and at the same time makes up for the shortcomings of a traditional literature review. Hiroshi Tanaka et al. [38] conducted an in-depth analysis of the scientific output of global desalination research from 1991 to 2008 through bibliometric methods and evaluated research trends. Wei Lun Ang et al. [39] evaluated the research trend of forward osmosis in seawater desalination and sewage treatment in the past 10 years with the method of bibliometrics, and focused on the future prospects and research directions of forward osmosis. Considering the value of seawater desalination, few studies have attempted to collect data on the results of global graphene-based membrane seawater desalination.

In this paper, a quantitative analysis of scholarly publications in the field of desalination based on graphene-based membranes is presented. The bibliometric approach was employed to analyze the literature reported between 2011 and 2022. Annual progression, research area, and geographic distribution, and keyword co-occurrence were examined. The objectives of this review were: (1) to quantitatively analyze the characteristics of the outputs of publications, country, subject categories, journals, and keywords about the topics; (2) to summarize research hotspots based on the analysis of the keywords co-occurrence network; (3) to focus on the preparation and application process of graphene-based membranes; and (4) to discuss the trending topics and future research directions.

## 2. Bibliometric Analysis Methodology

### 2.1. Data Source and Search Criteria

Data were retrieved from the Web of Science core collection database. The reason for choosing the Web of Science core collection database is the ease of gaining information for analysis. We used nearly a decade of publications to draw an image of what happened in the previous decade to predict what is most likely to be happening in the next decade. The literature related to graphene-based membranes for water desalination was retrieved with the following text format: Topic = (desalination) and (membrane) and (graphene) where topic search implies looking for searched terms or schemes in the title, abstract, author keywords, and keywords plus. Articles were based on the year from 2011 to 2022. The retrieved results can be found in the Appendix A. Keywords play an important role in extracting data as it confirms the accuracy of the data for the topic to analyze.

### 2.2. Analysis Method

The Bib-Excel and MS Excel 2019 software were used to sort the data. After sorting the data, the VosViewer software (version 1.6.13, Leiden University, Leiden, The Netherlands) was used for visual analysis [40,41]. Analysis was carried out based on the characteristics of the outputs of publications, country, journals, and the frequency of the used keywords, etc. The CiteSpace (version 5.7.R2, Drexel University, Philadelphia, PA, USA) was used to obtain a list of burst words [42,43,44,45]. The resulting bibliometric analysis was used to measure the publication’s contribution to knowledge advancement, future trends, author’s contributions, and research interests.

## 3. Bibliographic Analysis Results

### 3.1. Current Scenarios Regarding Publishing

There were 1739 articles related to desalination that used graphene materials and membranes in the Web of Science core collection database, including seven document types (Figure 1a). Journal articles (1518) comprised 85.67% of total production, followed by review papers (195, 11.21%). English was by far the dominant language. The year 2021 shows the highest number of publications (Figure 1b). A total of 356 articles were published in the year 2019, followed by 342 in the year 2020, 273 in 2019, 262 in 2018, and 153 in the recent months of 2022. Several publications for 2022 were up to July since 2022 was halfway, so the value would change at the end of this year.

### 3.2. Analysis of Leading Countries, Top Institutions and Funding Sponsors

All of these 1739 publications were from 69 countries. Countries that have published more than five papers were shown in Figure 2. Over the past 10 years, more and more countries have become increasingly involved in research in this area. The top 10 countries were ranked according to the highest number of publications, and the highest number of citations was shown in the Appendix A and Figure 3. Among the publication cases, China dominated with 870 publications, accounting for 50.00%, followed by the USA (18.05%), Iran (8.39%), Australia (7.36%), and South Korea (6.67%). It is worth mentioning that in terms of the number of citations, the total number of citations in China is 13,767, but the average citation rate is only 15.82%. Despite ranking first in the highest number of publications in China (Appendix A), India has the highest average citation rate. The possible reasons are as follows: (1) the work done by the Chinese authors has little influence or is not used in development; (2) China has been paying attention to this topic in recent years, so the new work has not received much attention.

Figure 3 depicts the cooperation network of the top 25 countries. The larger the node, the more the number of publications. The thickness of the line represents the number of cooperation, that is, the closeness of the cooperation relationship. Figure 3a illustrates the cooperation relationship between countries. The largest node was China, with the highest number of publications. There were links between almost all countries, indicating that international cooperation in this field was extensive. Most obviously, China has the most frequent exchanges with the USA, followed by Australia, Singapore, Belgium, and South Korea. From 2018 to 2020, China seemed to be very concerned about seawater desalination with graphene-based membrane, while the USA published the most articles before 2018. Australia, South Africa, Iran, and Canada, paid close attention to this field in 2019. Figure 3b displayed a diagram of the national publication distribution network divided by citation scores. In terms of citation scores, China and the USA showed the highest citation scores, indicating that the quality and influence of articles published may be higher.

The top 10 affiliated institutes and funding sponsors were shown in Appendix A. From the graphs, it is clear that China was highly funded and focused on this area of research. The highest number of publications were from the Chinese Academy of Science. Of the top 10 affiliations, five institutes were from China, which showed the level of seriousness of China to this research area. Due to collaboration between researchers and institutions, publications may be relevant to multiple institutions.

### 3.3. Publication Distribution by Journals and Subject Category

Appendix A shows the top 15 journals with the highest number of published articles. Desalination journals held top rank during the years 2011–2022 with 167 papers published, accounting for 9.70% of the total, and 7715 citations. The impact factor of desalination in 2021 was 11.21, followed by the Journal of Membrane Science (156 articles), Separation and Purification Technology (73 articles), and the Journal of Materials Chemistry A (66 articles). ACS Nano has the highest impact factor (18.02), and its publication number ranks 13. The IF of five journals exceeds 10, indicating that the field has a certain influence.

The Journal of Membrane Science seemed to have published the highest number of journals in 2020, but its publication volume declined in 2019 (Figure 4a). The number of papers published by Desalination and Separation and Purification Technology on graphene desalination has maintained an increasing trend, especially Separation and Purification Technology, which was currently more inclined to articles in this field, while Journal of Materials Chemistry A was exactly the opposite. In general, membrane technology was gaining popularity for its efficient separation and energy extraction at the same time, which was the reason it gained a rising trend this year.

Appendix A and Figure 4b show detailed analysis results of publications according to subject categories. There was considerable variety across 15 subject categories in publications. Only four categories accounted for more than 10%. There has been a significant rise in the research of materials for low energy and high recovery consideration. Engineering ranked first with 733 publications, accounting for 20.44% of the total publications, and chemistry ranked second with 678 kinds of publications, followed by materials science, science technology, other topics, and physics. Some papers, which fall under multiple categories of subjects, were counted in multiple subject categories. The statistical analysis results showed that graphene-based membrane desalination also involved biology and physics, indicating that this field involved interdisciplinary research, making the research more systematic and complete.

### 3.4. Analysis and Network of Author Keywords

As the author keyword analysis provides researchers with valuable knowledge on research trends, it has proven to be extremely useful for tracking research topic development. According to the results, from 2011 to 2021, there were 4750 total keywords, 2890 author keywords and 2288 index keywords. Out of 2890 author keywords, 2282 (79%) keywords appeared only once, showing the lack of research continuity and far differences in the research focuses. The top 15 most repetitive and active author keywords were listed in Appendix A. The keyword “desalination” frequently occurred 326 times in the publications, making up 11.28% of the author keywords. Further, graphene oxide appears 244 times (8.44%), graphene 119 times (4.11%), membrane 118 times (4.08%), reverse osmosis 84 times (2.90%), forward osmosis 83 times (2.87%), and so on. Other than desalination, graphene oxide, reverse osmosis, membrane, forward osmosis, graphene, and nanofiltration, were most frequently used keywords and observation proves that membrane technology was dominating the field of “desalination” in the trend.

The VOSViewer was used to generate keyword co-occurrence networks. The associations and importance (weights) of the 100 most relevant keywords are shown in Figure 5a,b. The research hotspots in graphene membrane seawater desalination in the past 10 years split into three clusters with different colors: the first group of studies focused on the applications of performance analysis in graphene oxide, forward osmosis, reverse osmosis, salt rejection, polyamide, and antifouling. The cluster appeared earlier and is a more mature cluster (Figure 5b); cluster 2 focuses on seawater desalination membrane research, including membrane modeling analysis, and sewage treatment, such as molecular simulation, graphene, carbon nanotubes, seawater desalination, reverse osmosis, and membranes; and the third category was related to the topic of solar desalination, such as “photothermal conversion”, “solar steam generation”, “solar desalination”, “water evaporation”, and “water purification”.

Based on Figure 5b, most keywords appeared in the past five years. The nodes in the graph gradually transition from the purple area to the yellow. The node color is yellow to indicate that the keyword is a new research topic. It is worth mentioning that some keywords have recently emerged, such as photothermal conversion, solar steam generation, membrane distillation, 2D materials, interlayer, membrane modification, and polyvinyl alcohol. It may reflect that the field is just getting started. Studies have shown that membrane methods were less energy-intensive than thermal methods. Energy consumption directly affects the cost-effectiveness and viability of desalination technology, such as reverse osmosis (RO), and reverse electrodialysis, which has drawn considerable interest lately. It was estimated that 63.7% of the total potential of the global desalted water was generated by membrane processes and 34.2% using thermal methods. However, thermal energy conversion utilizing sunlight may be a future research trend.

### 3.5. Burst Keywords Analysis

Burst words can reflect emerging or upcoming research topics in a research field [46]. The Citespace software was used to analyze burst words in graphene membrane desalination. The burst words between 2011 to 2022 are shown in Figure 6. The red bar indicates the period when the keyword burst. The dark blue bar indicates the time when the keyword appears. “Carbon nanotube membrane”, “reverse osmosis”, “water”, and “energy”, were the frontier fields that attracted the attention of early researchers. “Porous graphene” had the highest mutation intensity (15.57) in the four years after 2014, followed by “single layer graphene” (10.64), indicating a significant turning point in the development of this field. Burst words such as “dye”, “polydopamine”, ‘photothermal material” (heat-driven/solar-driven vapor generation), “film nanocomposite membrane”, and “highly efficient”, were prominent words in recent years. The dye represented two aspects of research: (1) dyes as cross-linkers to prepare composite membranes [47]; and (2) high-performance dye/dye separation and dye desalination applications [48,49,50]. Polydopamine was used as an anchor molecule leading to immobilization [51]. In addition, the development of new materials and heat-driven/solar-driven seawater desalination (Figure 7) will become an emerging research trend.

According to the literature assessment data, graphene-based membrane has become one of the important research areas of desalination. Graphene was considered as innovative material for desalination due to its tunable functions and atomic thickness advantages. This review paper will discuss the production of graphene and its derivatives. In addition, recent advances of graphene-based desalination membranes will be given to the three desalination processes: reverse osmosis, forward osmosis, and solar desalination.

## 4. Synthesis of Graphene-Based Membrane

In the mainstream seawater desalination process, graphene membranes are divided into three types: (1) single layer graphene is used as a barrier, and sub nano pores need to be made in graphene materials to allow mass transfer [52]; (2) the multilayer materials are realized by the size exclusion effect of the interlayer nanochannels and the gap between the nanosheets, electrostatic interaction, and ion adsorption, to achieve desalination [53]; and (3) graphene or its derivatives are loaded onto the polymer matrix, and the narrow gap between them acts as a water channel [54].

### 4.1. Synthesis of Graphene Membrane

Mechanical exfoliation and chemical vapor deposition are the two most common methods for preparing graphene [55,56]. The mechanical exfoliation method was the first method to be discovered to prepare graphene. Since the graphene sheets are combined with weak van der Waals force, graphene can be obtained by simply repeated exfoliation. However, mechanically exfoliated graphene has poor controllability and uniformity, making it impossible to achieve large-scale production and application. In chemical vapor deposition, carbon atoms are deposited and grown on the catalyst substrate by high-temperature cracking of the carbon source. The size of graphene is related to the substrate. The 30-inch continuous graphene films were firstly profoundly investigated by Sukang Bae et al. [57]. Based on these researches, F.M. Kafiah et al. [58] fabricated single-layer graphene using chemical vapor deposition and simultaneously transferred graphene onto a polymer microfiltration membrane for KCl ion removal. Nevertheless, methane is the most widely used carbon source in the preparation of graphene, and the proportion of carbon atoms converted into graphene varies by 1/10,000, which leads to high production costs. Subnanopores in graphene can be created and controlled by plasma etching or ion bombardment [59]. The first approach primarily used to construct such pores was electron beam penetration. Researchers have recently come up with an effective way to build the necessary pores on graphene sheets. A gallium ion pistol was used in this process and this device scans the graphene sheet from left to right and from top to bottom. An erratic gallium ion beam was fired at the graphene sheet allowing the gallium ions to be dispersed all over the graphene sheet creating distortion in the crystal lattice and knocking off carbon atoms. Subsequently, the defects produced will be prone to etching. Graphene was put within a solution of acidic potassium permanganate that was often used to eliminate carbon nanotubes. The outcome of using this approach in their work was a single sheet of graphene containing five trillion 0.4 nm holes/cm. However, for desalination purposes, experimental analysis indicates that the average diameter of the pore should be roughly about 7 angstroms [60,61]. Nano-sized pores were created in the graphene monolayer using an oxygen plasma etching process, which allows the tuning of the size of the pores [62]. The resulting membranes have near 100% salt rejection and fast water transport. Meanwhile, the single-layer graphene with nanopores deposited on the ultrafiltration membrane exhibited high water permeability and salt rejection [63,64,65]. Despite great superiorities, graphene nanoporous membranes still have shortcomings that challenge their expansion potential. It is still a great challenge to synthesize large-area defect-free sheets and produce pores with high density and uniform pore size.

### 4.2. Synthesis of Graphene Oxide Membrane

Graphene oxide (GO), a derivative of graphene, is often prepared by a modified Hummers method, which clears the price barrier for the practical application of GO thin films. GO has an ultrathin two-dimensional layered structure, and the nanochannels formed by the stacking of sheets can serve as channels for water molecules to pass freely, while blocking the passage of macromolecules whose molecular size is larger than the interlayer spacing. This enables selective filtration and separation applications of GO membranes. There are also a large number of oxygen-containing functional groups on the surface of GO, which make it have dispersibility in water or some organic solvents. Moreover, GO thin films can be prepared by some liquid-phase molding methods, such as the drop method, spin coating method, and vacuum filtration method, etc. Since, Nair and his collaborators [66] have reported in Science the preparation of GO films by spin coating. The results show that, for the GO film, many small molecular gases cannot penetrate the GO film, but water vapor can be transported freely, which is mainly attributed to the fast transport of water molecules in the nanochannels formed by the stacking of nanoclusters. GO-based filtration membranes have gradually become a global research hotspot. Sun et al. [67] prepared self-supporting GO films by the solution dropwise method and used a self-made filtration device to explore their mass transfer properties for different salt ions and organic dye molecules. Although the GO sheets can be held together by the interaction of hydrogen bonds and π-π bonds, good hydrophilicity causes the structure of the GO film to expand when it encounters water, which weakens the retention rate of the GO film. At the same time, benefiting from the abundant oxygen-containing functional groups on the surface, the structural stability of GO films can be improved by chemical modification. The lamellar spacing of GO films can be accurately adjusted to achieve different application purposes. In addition, selective filtration, and the separation performance of salt ions of different charges, can also be achieved by surface charge modification.

### 4.3. Synthesis of Reduced Graphene Oxide

To improve stability and salt rejection, GO nanosheets can be reduced to partially remove hydrophilic oxygen-containing groups, thereby increasing hydrophobicity and reducing interlayer spacing. GO is reduced by a chemical reaction to produce reduced graphene oxide (RGO). The RGO films with smaller lattice parameters (~0.34 nm) and graphene-like properties are ideal candidates, which could theoretically exclude blocking salt ions based on size. Annealing by rapidly heating GO results in the decomposition of some oxygen-containing functional groups into gases, creating sufficient pressure between the layers to separate [68]. Xiaoyi Chen et al. [20] prepared porous GO nanosheets by chemical etching using hydrogen peroxide, which was then deposited on porous support ultrafiltration membranes by vacuum filtration and then reduced by exposure to hydroiodic acid solution to generate RGO membranes. For the first time, the water permeability and Na_2_SO_4_ rejection of the RGO membrane reached the level of commercial polyamide-based nanofiltration membranes simultaneously. Hsin Hui Huang et al. [69] fabricated uniform RGO films by adjusting the hydrothermal reaction temperature and time through simple experiments.

## 5. Applications of Graphene-Based Membranes

### 5.1. Reverse Osmosis (RO)

To date, RO was recognized as the most efficient desalination technology for desalination on a large scale. RO based on polyamide (PA) thin film composite membranes has dominated the desalination field. However, its permeability was still compromised by chlorine during disinfection. The membrane was less tolerant to high temperature and pressure. Therefore, polymer membrane-based desalination needs further improvement, and its flux, antifouling properties, and chemical resistance, need more in-depth consideration. Eliminating the pretreatment process by developing anti-fouling membranes with surface properties that inhibit the attachment of debris will significantly reduce energy consumption and costs. The modification of the membrane with GO enhanced the antimicrobial performance and antifouling properties for RO. As an example, Cao et al. prepared PA membranes for RO using an interfacial polymerization method [70]. GO-modified thin-film polyamide membrane (ESPA-GO) was used to inhibit RO scaling in the presence of gypsum. The ESPA-GO membrane also strongly inhibited the gypsum scale compared to the unmodified ESPA membrane, reflecting the effect of its negative charge. GO nanosheet-loaded PA thin film composite (PA-TFC) membranes for RO were prepared by the layer-by-layer deposition technique [71]. The modified membrane exhibited high chemical stability, hydrophilicity, and water permeability. PA TFC membranes also exhibited strong resistance to chlorine- and fouling-induced membrane degradation. This reference applied TiO_2_ nanoparticles and GO, which self-assemble layer-by-layer onto the PA RO membrane surface to improve the membrane durability and made it more resistant to chlorine and anti-fouling [54]. Hee Joong Kim et al. modified the surface of graphene oxide (GO) with tannic acid (TA) [72]. The PA-GO-TA membrane exhibited excellent chlorine resistance (high flux of 34.21 L m^−2^ h^−1^ and 96.80% salt rejection) and good antibacterial properties during RO. It also exhibited excellent hydrophilicity, polymer matrix compatibility, and oxidative stress capacity. In addition, the rapid development of nano-enhanced reverse osmosis (RO) membrane showed the prospect of seawater desalination. Many theoretical studies have shown that nanoporous graphene has great potential applications in seawater desalination. Graphene was modified in this new membrane, by forming nanopores on the graphene surface. Throughout this process, well-structured pore channels can facilitate the flow of water and thus make the flow fast. Therefore, it was not only possible to remove salt using this approach, but also to filter other materials depending on their molecular size by changing the size of the pores. Physical rules such as charge and hydrophobicity were used to reject ions and other solvent molecules [60]. Despite its marginal thickness, graphene has high mechanical strength and these two properties help in low-pressure requirements and the quick transportation of water. The size of these pores needs to be precise. The puncturing process has to be carried out with special care and precision. Unless the pore sizes were small enough, it was difficult for the water molecules to enter the pores and desalination was impossible. When the pore sizes are too large, the salt molecules will flow through with the extremely undesirable water molecules.

One of the main challenges of graphene-based membranes for RO application is to simultaneously increase water permeability and the ability to reject salt. The salt rejection was found to be dependent on the membrane properties and operating conditions. It has been shown that boosting pore size and pressure can also weaken the membrane’s ability to resist salt. The water flow was directly proportional to the amounts of pores on the graphene sheet. However, by increasing the number of pores, the mechanical stability of the membrane would decrease. In addition, the distribution of pores was another issue of this process, as well as the mass processing of graphene sheets, which need to be solved by comprehensive research [60]. While promising results have been obtained for nanoporous graphene membrane in RO applications, it is noteworthy that most of the results obtained were based on simulation or laboratory tests [64,73,74]. Commercial applications of these novel membranes have not yet been obtained. The large-scale preparation of graphene sheets was also a challenge. Currently, the widely used chemical vapor deposition method is costly and produces mainly multilayer graphene films. Furthermore, it was difficult to generate regular nanopores on graphene sheets. The methods such as plasma etching and ion bombardment are expensive. Moreover, these techniques were immature and prone to introduce irregular nanopores, which lead to stress concentration and reduce the mechanical strength of graphene films. Further work and studies will make this process commercially viable and feasible. Due to the low pressure on the membrane, graphene-based membranes were mainly used in FO rather than RO.

### 5.2. Forward Osmosis (FO)

Forward osmosis (FO) is a relatively recent industrial method of desalination in which a gradient of the salt concentration (osmotic pressure) is the driving force through a synthetic membrane. On one side of the semi-permeable membrane is the feed (such as seawater), and on the other side is a higher osmotic pressure “draw” solution. Without applying any external pressure, the water coming from the feed solution would naturally move to the draw solution via the membrane. Then, the diluted solution is processed, separating the liquid from the reusable drawing solution. Therefore, the final solution of FO is not fresh water and requires secondary separation [62]. Yang et al. prepared single-layer graphene sheets by chemical vapor deposition, using O_2_ plasma to sequentially drill holes to produce nano-sized holes. Single-walled carbon nanotubes (SWNTs) were transferred on top of the graphene film, and the thus prepared graphene-nanomesh/SWNTs showed close to 100% rejection of salts (KCl, NaCl, and LiCl) in FO. This is the highest value reported using graphene. However, the desalination performance was evaluated by measuring the amount of infiltration and the conductivity of the water after 24 h. The relationship between penetration and time has not been considered. Therefore, the research only verifies the feasibility of this material in seawater desalination and lacks verification of its stability and practical application [65]. In another attempt, Peifu Cheng et al. [75] reported a new method for pore formation and interfacial polymerization for sub-nanoscale separation in large-area atomically thin graphene films. Graphene grown at CVD~900°C uses mild etching conditions with UV/ozone exposure to enlarge existing defects in graphene, and introduces additional nanopores in the graphene lattice. Finally, the tears and large nanopores in graphene are selectively sealed. Scalable centimeter-scale nanoporous graphene membranes were used, exhibiting excellent rejection of salt ions (97%) and small-molecule rejection ~100% in FO.

For GO, the most critical issue is the swelling of GO in an ionic solution, which limits their sieving of small-sized ions. Therefore, recent research has focused on addressing this issue. Sun et al. explored a GO/monolayer TiO_2_ hybrid membrane with excellent static desalination performance by embedding monolayer TiO_2_ nanosheets into GO sheets by vacuum filtration [76]. The GO was gently reduced to RGO by the photocatalytic ability of TiO_2_ under UV irradiation. The intercalation of TiO_2_ and the reduction of oxygen-containing functional groups reduced the interlayer spacing, thereby increasing ionic repulsion. Compared with GO/TiO_2_, the RGO/TiO_2_ laminated film can reduce ion permeation by 95%, while retaining 60% of water transport. Furthermore, the lack of stability made this membrane limiting for use conditions, considering that the TiO_2_ layers were embedded in the stacked GO laminates via the vacuum filtration method.

Filtration of wastewater through membranes causes microbial cells to attach to the membrane surface. The presence of biofilms can impair membrane performance. Due to the antibacterial activity and low toxicity to cells of carbon-based nanomaterials (CNT and GO), Perreault et al. [77] used a simple amide coupling (carbodiimide chemistry) between the carboxyl groups of GO and the carboxyl groups of a polyamide (PA) layer. Bacterial exposure to the GO-functionalized membrane resulted in 65% bacterial inactivation after 1 h, while other transport properties of the functional membrane were not adversely affected. Hegab et al. [78] grafted GO to a thin film composite (TFC) PA via poly l-Lysine (GO/PLL-LbL) or a hybrid (H) grafting strategy (GO/PLL-H) to enhance the antibacterial properties of FO. The GO/PLL-LBL tuned surface exhibited higher hydrophilicity and smoothness, and a 99% reduction in viable bacteria. In addition, the hybrid modification (GO/PLL-H) also improved the selectivity of membrane salts compared to the pristine membrane. Therefore, a simple and stable modified FO membrane can control and mitigate membrane biofouling for long-lasting performance, thus meeting the requirements of practical applications. However, it must be remembered that when desalinating brackish water or brine, the flux rate will be greatly reduced compared to the values obtained from experiments using pure water feed, and the performance will also depend on the choice of scheme. Therefore, the desalination efficiency and stability of the actual feedwater should be considered in the design of graphene membranes in the future.

### 5.3. Solar Desalination

As one of the cleanest renewable energy sources, solar energy could provide a promising solution. The photothermal effect is the driving force of the natural hydrological cycle and atmospheric circulation, and it brings inspiration to solve the problem of water shortage. The sunlight, as the only energy source, can evaporate water through the photothermal effect, breaking through the huge bottleneck between freshwater production and energy consumption. However, the traditional solar-driven evaporation system has bottlenecks such as poor light absorption efficiency and fast heat dissipation rate. In this case, researchers have spent a lot of effort developing advanced and efficient solar-driven evaporation-based desalination technologies to take advantage of sunlight. The three major components that make up a solar interfacial evaporation system are photothermal materials, heat management (insulation to reduce heat loss), and water supply (essential to maintain evaporation). Among them, photothermal materials play a significant role in the final conversion efficiency. To date, various carbon-based materials have been made into excellent candidates for photothermal materials. It can absorb part of the photoelectromagnetic spectrum of visible light and near-infrared, and can also absorb radiation into heat through a non-radiative decay process with excellent photothermal conductivity, including CNTs, graphene, carbon nanotubes, carbon nanotubes, and carbon nanotubes, GO, RGO, and carbon black, etc. In this section, we present some of the interesting research into graphene material for solar desalination. Li et al. [7] designed a portable desalination device (see Figure 8). The fabricated foldable graphene oxide thin-film devices can achieve minimal heat loss, while the energy transfer efficiency is independent of the amount of water and can be maintained without container insulation.

In terms of efficiency, the vertical design of the solar evaporator architecture is superior to the flat configuration. Shenghao Wu et al. [79] recently reported the composition of vertically oriented graphene nanosheets (VGs) directly in-grown on the surface of graphene aerogels (GA) by plasma-enhanced chemical vapor deposition (PECVD). The schematic of VG/GA is shown in Figure 9a–f. An ultrafast solar thermal response (169.7 °C temperature increase in 1 s) was achieved; this material has excellent photon absorption and excellent thermal insulation. The high energy efficiency of saturated steam was obtained at 10 suns (89.4%). Importantly, it demonstrated the high-throughput, scalable fabrication of unique nanostructures. Other researchers have attempted to produce clean water by preparing long-range vertically aligned sheets membrane (VA-GSM) as high-efficiency solar-thermal converters (Figure 9g). VA-GSM achieves average water evaporation rates of 1.62 and 6.25 kg m^–2^ h^–1^ under 1-day and 4-day illumination, and the solar-to-heat conversion efficiencies were as high as 86.5% and 94.2%, respectively. The performance of VA-GSM outperforms most of the previously reported carbon materials and can efficiently generate clean water from seawater, ordinary wastewater, and even concentrated water [80].

Nature is the greatest creator of impressive structures, inspiring structural design in many fields of study. For example, plants often have multi-channel structures that transport water and nutrients from the roots to the upper trunk, which are used during photosynthesis. The natural structure of plants inspires various biomimetic designs for high-performance solar evaporation devices. Ostrikov and colleagues deposited vertically oriented graphene (VG) on a lotus microtextured surface (lotus/VG) by one-step plasma treatment of the holy lotus (*Nelumbo nucifera*) (Figure 10a,b). It demonstrated an interesting integrated structure using natural materials [81]. The innate porous structure of the lotus was combined with a complementary nanosheet coating, which facilitates better light trapping, resulting in a broadband light absorption of up to 99.2%. Natural materials were generally regarded as promising light absorbers, which are abundant and low-cost. Likewise, natural solar evaporator materials have great potential for future large-scale applications. In addition, Xia et al. [82] reported a solar steam generator similar to the shape of a lotus leaf, which consists of a horizontal evaporation disk and vertical suction lines (Figure 10c). The filter paper was used as a water-spreading layer. CNT acted as a light-absorbing layer and cotton thread acted as water uptake thread. Its design causes the salt to crystallize only at the edges of the evaporation pan. By weakening the bond between the salt crystals and the evaporating disk, the salt automatically falls off under the force of gravity. Due to the edge-preferred crystallization and self-cleaning properties, the solar steam generator achieves uninterrupted long-term operation (over 600 h uninterrupted) to continuously generate steam and harvest salt [82]. Recently, changing the structure of the absorber from traditional 2D to macroscale 3D structure, provides a new way to improve the performance of solar steam beyond current solar energy. Another study [83] also conducted related research and proposed a strategy for the synchronization of self-acting desalination and evaporation. 3D T-shaped porous sponges (3DTPS) were designed for various 2D photothermal thin films to simultaneously achieve self-acting desalination and additional evaporation (Figure 10d). Ti_3_C_2_T_x_ and polydopamine were used as photothermal materials for 2D photothermal films. 3DTPS provides water exchange channels for 2D photothermal films. Desalination can also be maintained under one sun (1 kW m^−2^) when the brine concentration was as high as 20 wt%. 3DTPS (D = 2.5 cm) was used as an additional evaporation structure, resulting in a more than 21% increase in evaporation efficiency.

In addition to 2D materials, researchers showed that 3D materials also have good properties. The vertically aligned graphene pillar array (HOPGF) has an enlarged evaporation area and additional free space for fast vapor escape. It has an extremely high evaporation rate of 2.10 kg m^−2^ h^−1^ under only 1 sun, which was the best among carbon-based materials [84]. RGO nanosheets, straw-derived cellulose fibers, and sodium alginate (SA), were fabricated into 3D photothermal aerogels for solar steam power generation. During solar steam power generation, the 3D photothermal aerogel effectively reduces radiation and energy losses, while enhancing energy harvesting from the environment. It has an extremely high evaporation rate of 2.25 kg m^–2^ h^–1^, corresponding to an energy conversion efficiency of 88.9% under 1.0 solar irradiation, while the salinity of clean water collected during the evaporation of real seawater was only 0.37 ppm [85]. In addition to graphene alone, combining plasmonic resonance with graphene can achieve unexpected properties. Through careful design, Xu et al. prepared an N-doped graphene sea urchin photothermal material embedded with copper nanodots, which was achieved by nanoscale plasmonic resonance, nonradiative relaxation of photoexcited electrons, and thermal vibration of molecules or lattices; simultaneously coupling the advantages of graphene to maximize solar energy absorption (>99%) across the entire solar spectrum. At the same time, a solar-powered desalination system was also manufactured, which has a desalination efficiency of up to 82%. This system produces fresh water at a rate of approximately achieving a water evaporation rate of ~0.63 kg m^−2^ h^−1^, namely ~5 L m^−2^ day^−1^ [86]. Wood was considered to be a good thermal insulation material with the characteristics of large soil storage, hydrophilic surface and high-water transfer efficiency. The incorporation of mesoporous three-dimensional graphene networks (3DGNs) with wood was shown to greatly enhance the solar-to-steam conversion efficiency (Figure 11). 3DGN deposited on wood provided excellent solar-to-steam conversion efficiency values of approximately 91.8% under one sun exposure [87].

The importance of the carrier was also studied to achieve good results by loading the material on a lightweight carrier. Fathi S. Awad et al. employed a plasmonic graphene polyurethane (PGPU) nanocomposite with metal nanoparticles (0.2 wt%) incorporated into 0.5 wt% GO-polyurethane (GOPU) nanocomposite (Figure 12). It absorbed sunlight more efficiently than bulk metal or carbon materials. The Au/Ag-PGPU foams exhibited average water evaporation rates of 11.34 kg m^–2^ h^–1^ with superb solar thermal efficiencies of up to 96.5% under 8 sun illumination. Furthermore, the PGPU foam showed stable evaporation rates over 10 repeated evaporation cycles without any performance degradation [88].

A study reported a plasmonic copper oxide/RGO polyurethane composite (2% Ag/CuO-rGO-PU) based on silver nanoparticles exhibiting very strong solar energy absorption. rGO and CuO nanoparticles have excellent heat absorption. Polyurethane was used as a carrier and thermal insulator. The Ag/CuO-RGO-PU film has an average of 2.6 kg m^−2^ h^−1^ under one solar irradiation, and the evaporation rate and solar thermal efficiency are as high as 92.5% [89]. The absence of a power source and the potential portability of the device represents an attractive solution for long-term desalination, and these studies are of great value in advancing our understanding of efficient photothermal conversion.

## 6. Conclusions and Perspectives

In this literature review, 1739 publications, published between 2011–2021, from 350 journals, were analyzed. The major contribution and citation to these publications were from China and the USA. There was a wide range of cooperative relations among all countries, and China and the USA cooperated most frequently. Some burst words of desalination were addressed and provide a perspective of the research trend over 10 years. Among them, “dye”, “polydopamine”, “photothermal material”, “film nanocomposite membrane”, and “highly efficient”, were the burst words emerging in recent years. The development of new materials and heat-driven/solar-driven seawater desalination are trending research topics. The latest developments of graphene-based desalination membranes were reviewed in the three desalination processes: reverse osmosis, forward osmosis, and solar desalination. Based on the literature analysis results, current research characteristics were clarified and the future research directions were enlightened.

To further accelerate the development, researchers should pay close attention to the following aspects:(1)The design of novel multifunctional nanocomposite membranes is always a focus. For example, the membranes with long-term stability are worthy of further investigation. The antifouling (organic, inorganic, and biological fouling) performance of the membrane is related to the hydrophilicity, roughness, charge density, and functional groups of the membrane. Therefore, graphene-based membranes should integrate antibacterial nanomaterials and the membrane surface should be functionalized. In addition, there are many different components of sea water. The molecular size, physical, and chemical properties of these components are different. These complex environmental applications present different challenges to the membranes used.(2)The application of graphene-based membranes has been limited to the laboratory scale, and the successful commercialization of large-scale industrial applications is still rare. Several criteria must be considered to bring membrane desalination into practical applications. Firstly, a low-cost method is needed to develop high-efficiency graphene-based membranes with excellent performance. Secondly, techno-economic analysis should be conducted to calculate the total cost of the desalination industry. Thirdly, the design and manufacturing process needs to be improved for the large-scale production of graphene-based desalination devices with high performance but low-cost. The above three points ensure its implementation in large-scale, practical, and practical field applications.(3)Seawater desalination is an energy-intensive process, and utilizing a hybrid system of renewable energy sources is proposed as a promising solution. Heat-driven/solar-driven desalination is regarded as a good strategy for a desalination system. Solar energy is a renewable and clean energy source and is considered a promising technology to provide a sustainable energy supply for desalination. In particular, graphene, as light absorption, and a light-to-heat conversion material, is an ideal candidate for solar-driven seawater desalination. Future studies should focus on how to further increase the evaporation rate. Firstly, innovative photothermal materials should to be designed and the structural design should be optimized, because the transport of water is closely related to the pore structure and surface properties. Secondly, the mechanisms of mass transfer, convection, and radiation, in membrane desalination processes need to be further unveiled. Simulation models are powerful tools for gaining insight into the fundamental behavior and dynamics of processes related to light, water, steam, and heat. Thirdly, an exploration of all-weather desalination should be conducted. Nearly all efforts with the current system are focused on laboratory testing, but light intensity varies with environmental conditions, such as air humidity, and rainy weather, etc. Possibly steam generation will be reduced to zero in dark conditions. Through long-term outdoor testing, researchers can better understand the gap between current systems and real-world applications. These future efforts will make graphene-based membranes a potential solution to water scarcity.

## Figures and Tables

**Figure 1 polymers-14-04246-f001:**
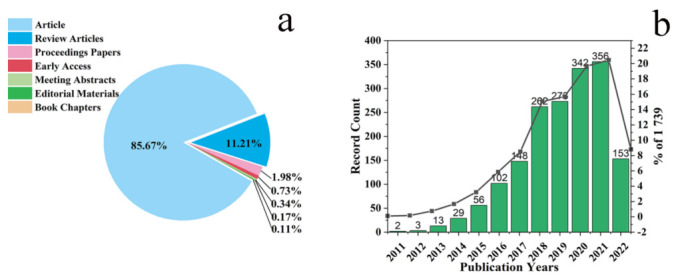
(**a**) Documents by type used for bibliometric analysis; (**b**) publications per year.

**Figure 2 polymers-14-04246-f002:**
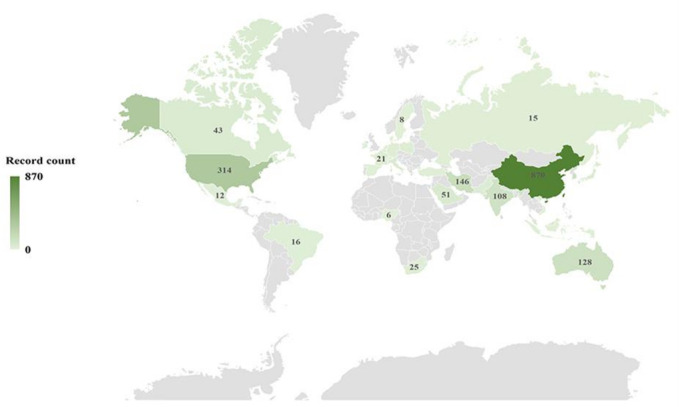
Worldwide geographical distribution of the graphene-based membranes on desalination.

**Figure 3 polymers-14-04246-f003:**
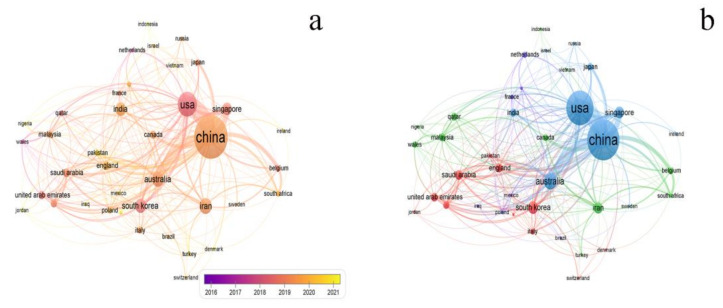
(**a**) The network of the top 40 countries. Each country on the map was displayed as a node, with size determined by the number of publications. Country relationships were shown as edges of varying thickness determined by the co-occurrence. The node color was determined by the publication time; (**b**) the network of publication distribution by countries in terms of citation scores. The node size was determined by the citation scores.

**Figure 4 polymers-14-04246-f004:**
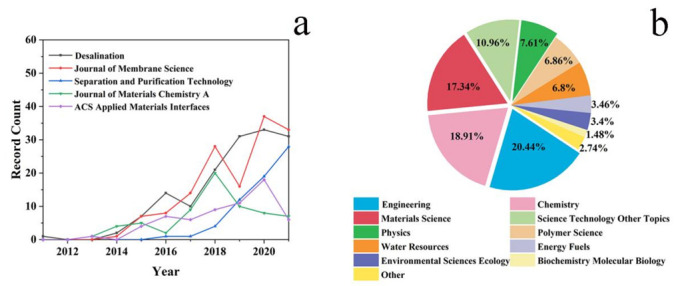
(**a**) Publications by year by top publication journals; (**b**) publications by subject category.

**Figure 5 polymers-14-04246-f005:**
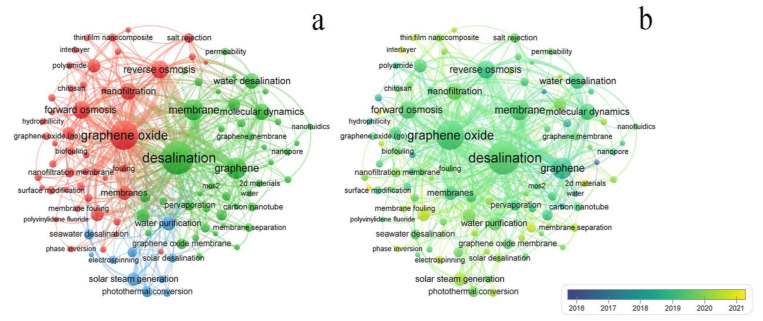
Density plot of author keywords: (**a**) network of author keywords; and (**b**) overlay graph of author keywords colored by the time.

**Figure 6 polymers-14-04246-f006:**
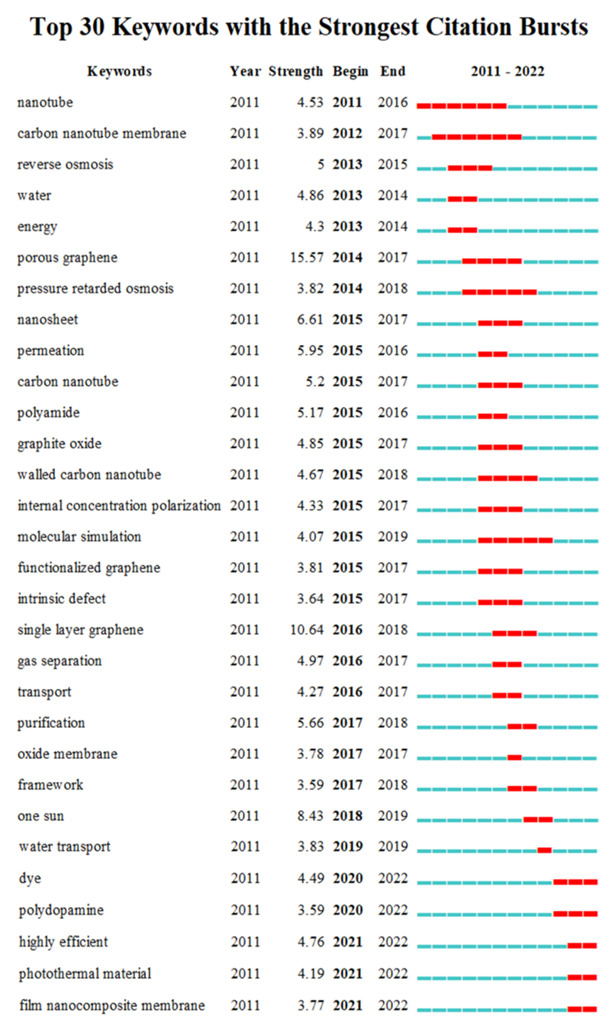
Burst keywords analysis based on publications from 2011 to 2022. The red represents the burst time.

**Figure 7 polymers-14-04246-f007:**
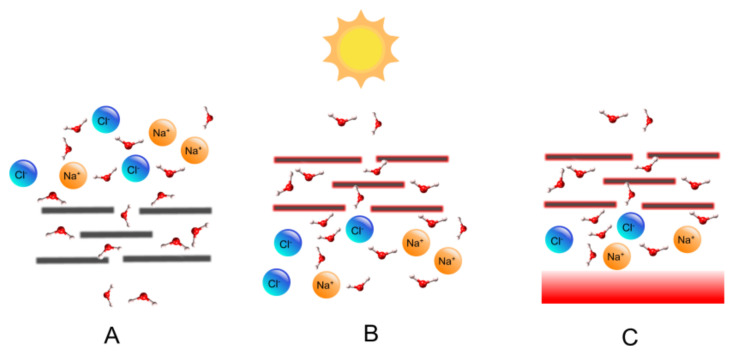
Schematics of: (**A**) typical filtration desalination; (**B**) solar-driven desalination; and (**C**) heat-driven desalination.

**Figure 8 polymers-14-04246-f008:**
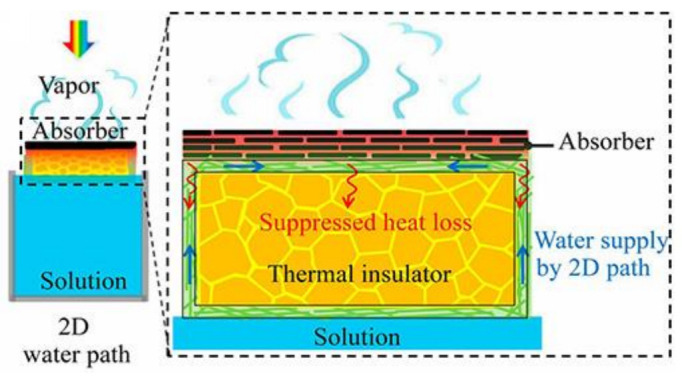
Schematic diagram of solar desalination device for 2D water supply [7]. Reproduced with permission. Copyright 2016, National Academy of Sciences.

**Figure 9 polymers-14-04246-f009:**
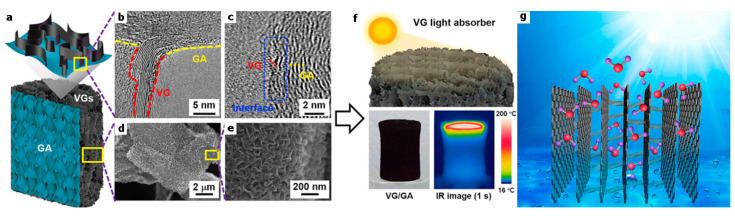
(**a**) Schematic of VG/GA; (**b**,**c**) high-resolution TEM images of VG/GA; (**d**,**e**) SEM images of VG/GA; (**f**) IR image of VG/GA [79] Reproduced with permission. Copyright 2019, ELSEVIER; and (**g**) schematic illustration of VA-GSM [80] Reproduced with permission. Copyright, American Chemical Society.

**Figure 10 polymers-14-04246-f010:**
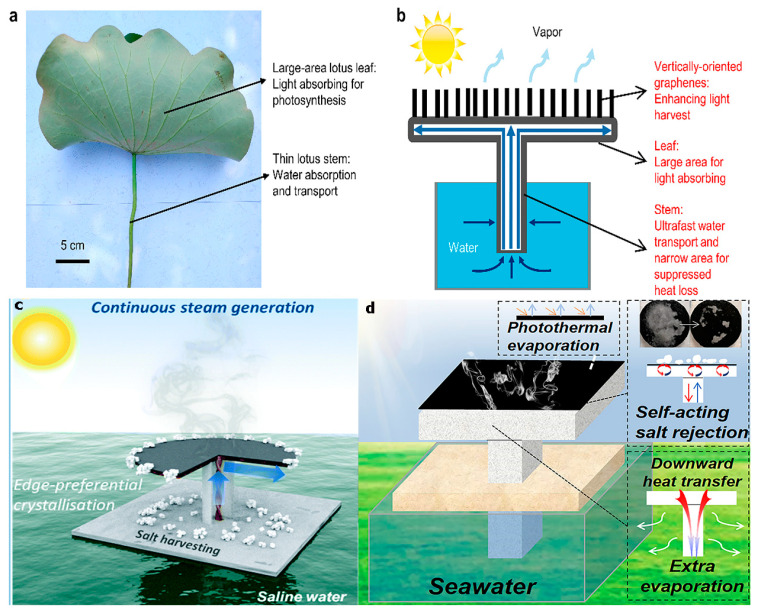
(**a**) Optical image of natural lotus; (**b**) schematic diagram of the lotus-based energy and water conversion and use system [81] Reproduced with permission. Copyright 2019, ELSEVIER; (**c**) design sketch of the solar steam generator desalination process [82] Reproduced with permission. Copyright 2019, Royal Society of Chemistry; and (**d**) structure of a solar-driven interfacial steam power generation system based on a 3D T-shaped porous sponge [83] Reproduced with permission. Copyright 2021, ELSEVIER.

**Figure 11 polymers-14-04246-f011:**
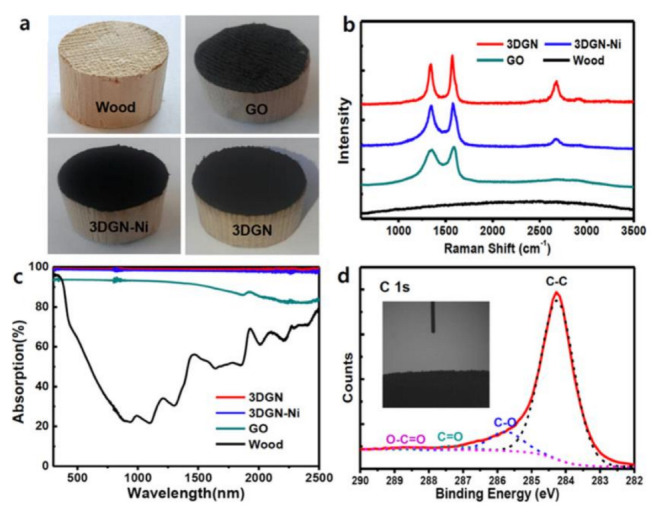
(**a**) DC image of the bare wood and GO, 3DGN-Ni, and 3DGN, on the wood; (**b**) Raman spectra; (**c**) UV–vis−NIR spectroscopy of wood, GO, 3DGN-Ni, and 3DGN; and (**d**) C 1s XPS spectrum of 3DGN [87] Reproduced with permission. Copyright 2018, American Chemical Society.

**Figure 12 polymers-14-04246-f012:**
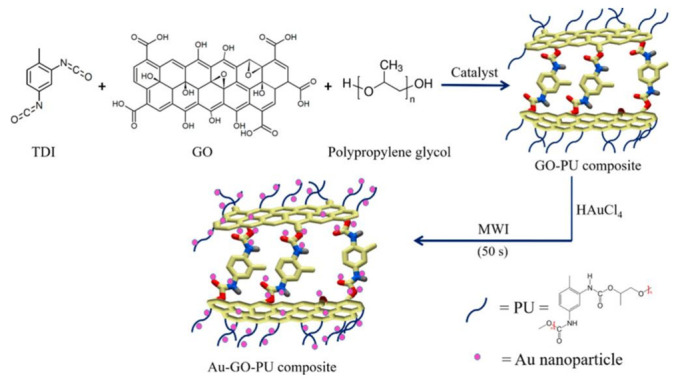
Design steps for the synthesis of PGPU nanocomposite foams [88] Reproduced with permission. Copyright 2017, American Chemical Society.

## Data Availability

The data presented in this study are available upon request from the corresponding authors.

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
