# Peer review of "Graphene-Based Membranes for Water Desalination: A Literature Review and Content Analysis"

_polymers, 2022, doi:10.3390/polym14194246_

Round 1

Reviewer 1 Report

This review is not well studied and organized to show the significant contribution for scientific findings or other practical engineering application. 

Granphene is well studied with lots of paper being published year by year, while its main scientific contributions and application seems not to be the water desalination as compared to the well used polyamide membrane with very commercial and stable technologies. 

As thus, how to justify the rationale and necessity for this research direction or topic with meaningful applications?

Author Response

Thank you for the comment. We sincerely hope that this review can provide clues and inspirations for the exploration of new desalination membranes in the future, and further promote the synthesis and large-scale production of desalination membranes. In this paper, our vision is to take an exhaustive examination of current research trends focusing on graphene and its derivatives to considerably improved performance of membrane technologies. Bibliometrics combines traditional literature review to better understand the research trend of this topic.

Graphene is well studied with lots of paper being published year by year, and some of the research has focused on desalination. Although commercialized polyamide membranes appear to be ripe for general application in RO desalination, optimal design is still required. The ideal membrane should have the highest possible water flux per operating pressure, near complete salt rejection, high fouling resistance and oxidant resistance. Special attention is paid to maximizing the water flux while maintaining the desalination rate, because the high-water flux of the membrane module will effectively reduce the energy cost under the same operating conditions. As a result, researchers around the world are constantly looking for strategies to maximize further desalination performance. On the one hand, RO membranes based on polyamide membranes have been extensively studied and modified to maximize their desalination performance. Various tailoring methods (grafting, deposition and merging) and support layers (deposition and merging) have been widely reported. Alternatively, new desalination membranes have emerged, some of which have shown great potential. For example, advances in nanotechnology have inspired the design of novel membranes based on two-dimensional (2D) nanomaterials such as graphene and its derivatives, which have greatly stimulated innovation in filter materials. Graphene-based desalination membranes are divided into three main types: 1. Single layer graphene is used as a barrier, and sub nano pores need to be made in graphene materials to allow mass transfer; 2. The multilayer mate-rials is realized by the size exclusion effect of the interlayer nano channels and the gap between the nano sheets, electrostatic interaction and ion adsorption to achieve desalination; 3. Graphene or its derivatives are loaded onto the polymer matrix, and the narrow gap between them acts as a water channel. The polymer film was used as the substrate in most studies, and the polymer film was not completely abandoned. In addition, graphene-based membranes have great advantages compared to commercial polyamide membranes, such as nanoporous graphene membranes with 2-3 orders of magnitude higher water flux than conventional RO membranes, and salt rejections close to 100%. Furthermore, the high surface hydrophilicity of GO filtration membranes is always beneficial to mitigate membrane fouling, especially biofouling. Energy consumption and costs will be significantly reduced by developing anti-fouling membranes with surface properties that inhibit the adhesion of impurities. Apart from this, graphene and derived materials possess other interesting properties, such as high thermal conductivity of about 5000 W m−1K−1, making them ideal candidates for solar-driven photothermal desalination processes. And have good electrical conductivity, which means they can be used as electrodes for capacitive deionization of seawater. Therefore, this research direction is necessary.

We have added the rationale and necessity of graphene-based membranes for desalination in the manuscript. The details are as follows:

The general application of polyamide membrane-based RO desalination seems to be well established, but it is worth noting that the most ideal membrane should have the highest possible water flux, complete salt rejection, high anti-fouling and oxidation tolerance. Therefore, it is still an urgent task for researchers around the world to explore more advanced seawater desalination materials and processes to further im-prove the performance of seawater desalination to the maximum extent. On the one hand, RO membranes based on polyamide membranes have been extensively studied and modified to maximize their desalination performance. Various surface modifications, including grafting, deposition and merging, have been widely reported [20, 21]. Alternatively, new desalination membranes have emerged, some of which have shown great potential. For example, carbon materials [22], molybdenum disulfide (MoS2) [23] and MXene membranes [24] show great potential in seawater desalination.

With its unique two-dimensional structure and fast mass transfer channels, graphene materials have received extensive attention for their applications in membrane separation such as water purification, solvent dehydration and gas separation [25, 26]. Despite some limitations, graphene-based membranes have great advantages compared to commercial polyamide membranes, such as nanoporous graphene membranes with 2-3 orders of magnitude higher water flux than conventional RO membranes and salt rejections close to 100% [27]. In addition, the high surface hydrophilicity of GO membranes is beneficial to mitigate membrane fouling, especially bio-logical fouling [28]. Energy consumption and costs will be significantly reduced by developing anti-fouling membranes with surface properties that inhibit the adhesion of impurities. Apart from this, graphene and derived materials possess other interesting properties, such as high thermal conductivity of about 5000 W m−1K−1, making them ideal candidates for solar-driven photothermal desalination processes [29]. They have good electrical conductivity, which means can be used as electrodes for capacitive de-ionization of seawater [30].

References

  1. Chen, X.Y.; Feng, Z.H.; Gohil, J.; Stafford, C.M.; Dai, N.; Huang, L.; Lin, H.Q. Reduced holey graphene oxide membranes for desalination with improved water permeance. Acs Applied Materials and Interfaces 2020, 12, 1387-1394.
  2. Chung, Y.T.; Mahmoudi, E.; Mohammad, A.W.; Benamor, A.; Johnson, D.; Hilal, N. Development of polysulfone-nanohybrid membranes using ZnO-GO composite for enhanced antifouling and antibacterial control. Desalination 2017, 402, 123-132.
  3. Manawi, Y.; Kochkodan, V.; Hussein, M.A.; Khaleel, M.A.; Khraisheh, M.; Hilal, N. Can carbon-based nanomaterials revolu-tionize membrane fabrication for water treatment and desalination? Desalination 2016, 391, 69-88.
  4. Heiranian, M.; Farimani, A.B.; Aluru, N.R. Water desalination with a single-layer MoS2 nanopore. Nat Commun 2015, 6.
  5. Zhang, B.P.; Gu, Q.F.; Wang, C.; Gao, Q.L.; Guo, J.X.; Wong, P.W.; Liu, C.T.; An, A.K. Self-assembled hydrophobic/hydrophilic porphyrin-Ti3C2Tx MXene janus membrane for dual-functional enabled photothermal desalination, Acs Applied Materials & Interfaces 2021, 13, 3762-3770.
  6. Novoselov, K.S.; Geim, A.K.; Morozov, S.V.; Jiang, D.; Zhang, Y.; Dubonos, S.V.; Grigorieva, I.V.; Firsov, A.A. Electric field effect in atomically thin carbon films. Science 2004, 306, 666-669.
  7. Allen, M.J.; Tung, V.C., Kaner, R.B. Honeycomb carbon: a review of graphene. Chemical Reviews 2010, 110, 132-145.
  8. Cohen-Tanugi, D.; Grossman, J.C. Water permeability of nanoporous graphene at realistic pressures for reverse osmosis de-salination. Journal of Chemical Physics 2014, 141.
  9. Matshetshe, K.; Sikhwivhilu, K.; Ndlovu, G.; Tetyana, P.; Moloto, N., Tetana, Z. Antifouling and antibacterial beta-cyclodextrin decorated graphene oxide/polyamide thin-film nanocomposite reverse osmosis membranes for desalination applications. Sep Purif Technol 2022, 278.
  10. Mahmoud, K.A.; Mansoor, B.; Mansour, A.; Khraisheh, M. Functional graphene nanosheets: the next generation membranes for water desalination. Desalination 2015, 356, 208-225.
  11. Leong, Z.Y.; Lu, G.; Yang, H.Y. Three-dimensional graphene oxide and polyvinyl alcohol composites as structured activated carbons for capacitive desalination. Desalination 2019, 451, 172-181.

Reviewer 2 Report

The paper by Dai et al is a mini-review on graphene-based membranes for water desalination. A huge work of data analysis has been made from the authors and for this reason, the paper is worthy of consideration. However, given the current  structure of the paper i.e. after a  broad introduction/section on data analysis, there is directly the section on the applications,  my concern is about the absence of a  section on the materials, necessary for this journal. 

Polymers journal focuses on the materials AND also on their applications, but surely  not ONLY on the applications, otherwhise its name should be for example PROCESSES, DESALINATION and so on.

Author Response

Thank you for the comment. Based on your suggestion, we have added an overview of the Materials section as follows:

  1. Synthesis of graphene-based membrane

In the mainstream seawater desalination process, graphene membrane can be di-vided into three types: 1. Single layer graphene is used as a barrier, and sub nano pores need to be made in graphene materials to allow mass transfer[51]; 2. The multilayer mate-rials is realized by the size exclusion effect of the interlayer nano channels and the gap between the nano sheets, electrostatic interaction and ion adsorption to achieve desalination[48]; 3. Graphene or its derivatives are loaded onto the polymer matrix, and the narrow gap between them acts as a water channel[61].

4.1. Synthesis of graphene membrane

Mechanical exfoliation and chemical vapor deposition are the two most common methods for preparing graphene [62, 63]. The mechanical exfoliation method is the first method to be discovered to prepare graphene. Since the graphene sheets are combined with weak van der Waals force, graphene can be obtained by simple repeated exfoliation. However, mechanically exfoliated graphene has poor controllability and uniformity, making it impossible to achieve large-scale production and application. In chemical vapor deposition, carbon atoms are deposited and grown on the catalyst substrate by high temperature cracking of the carbon source. The size of graphene is related to the substrate. 30-inch continuous graphene film were firstly profoundly investigated by Sukang Bae et al [64]. Based on these researches, F.M. Kafiah et al. [65] fabricated single-layer graphene using chemical vapor deposition and simultaneously transferred graphene onto a polymer microfiltration membrane for KCl ion removal. Nevertheless, methane is the most widely used carbon source in the preparation of graphene, and the proportion of carbon atoms converted into graphene varies by 1/10,000, which leads to high production costs. Subnanopores in graphene can be created and controlled by plasma etching or ion bombardment [66]. The first approach primarily used to construct such pores was electron beam penetration. Researchers have recently come up with an effective way to build the necessary pores on graphene sheets. A gallium ion pistol was used in this process and this device scans the graphene sheet from left to right and from top to bottom. An erratic gallium ion beam was fired at the graphene sheet allowing the gallium ions to be dispersed all over the graphene sheet creating distortion in the crystal lattice and knocking off carbon atoms. Subsequently, the defects produced will be prone to etching. Graphene was put within a solution of acidic potassium permanganate that was often used to eliminate carbon nanotubes. The outcome of using this approach in their work was a single sheet of graphene containing five trillion 0.4 nm holes/cm. But for desalination purposes, experimental analysis indicates that the average diameter of the pore should be roughly about 7 angstroms [67, 68]. Nano-sized pores were created in the graphene monolayer using an oxygen plasma etching process, which allows tuning of the size of the pores [47]. The resulting membranes have near 100% salt rejection and fast water transport. Meanwhile, the single-layer graphene with nanopores deposited on the ultrafiltration membrane exhibited high water permeability and salt rejection [54, 69, 70]. Despite great superiorities, graphene nanoporous membranes still have some shortcomings that challenge its expansion potential. It is still a great challenge to synthesize large-area defect-free sheets and producing pores with high density and uniform pore size.

4.2. Synthesis of graphene oxide membrane

Graphene oxide (GO), a derivative of graphene, is often prepared by a modified Hummers method, which clears the price barrier for the practical application of GO thin films. GO has an ultrathin two-dimensional layered structure, and the nanochannels formed by the stacking of sheets can serve as channels for water molecules to pass freely, while blocking the passage of macromolecules whose molecular size is larger than the interlayer spacing. This enables selective filtration and separation ap-plications of GO membranes. There are also a large number of oxygen-containing functional groups on the surface of GO, which make it have dispersibility in water or some organic solvents. Moreover, GO thin films can be prepared by some liquid-phase molding methods, such as drop method, spin coating method, vacuum filtration method, etc. Since Nair and his collaborators [71] reported in Science the preparation of GO films by spin coating. The results show that, for the GO film, many small molecular gases cannot penetrate the GO film, but water vapor can be transported freely, which is mainly attributed to the fast transport of water molecules in the nanochannels formed by the stacking of nanoclusters. GO-based filtration membranes have gradually become a global research hotspot. Sun et al. [72] prepared self-supporting GO films by solution dropwise method and used a self-made filtration device to explore their mass transfer properties for different salt ions and organic dye molecules. Although the GO sheets can be held together by the interaction of hydrogen bonds and π-π bonds, the good hydrophilicity causes the structure of the GO film to expand when it encounters water, which weakens the retention rate of the GO film. At the same time, benefiting from the abundant oxy-gen-containing functional groups on the surface, the structural stability of GO films can be improved by chemical modification and modification, and the lamellar spacing of GO films can be accurately adjusted to achieve different application purposes. In addition, selective filtration and separation performance of salt ions of different charges can also be achieved by surface charge modification.

4.3. Synthesis of reduced graphene oxide

To improve stability and salt rejection, GO nanosheets can be reduced to partially remove hydrophilic oxygen-containing groups, thereby increasing hydrophobicity and reducing interlayer spacing. GO is reduced by a chemical reaction to produce reduced graphene oxide (rGO). The rGO films with smaller lattice parameters (~0.34 nm) and graphene-like properties are ideal candidates, which could theoretically exclude blocking salt ions based on size. Annealing by rapidly heating GO results in the de-composition of some oxygen-containing functional groups into gases, creating sufficient pressure between the layers to separate [73]. Xiaoyi Chen et al. [20] prepared porous GO nanosheets by chemical etching using hydrogen peroxide, which were then deposited on porous support ultrafiltration membranes by vacuum filtration, and then reduced by exposure to hydroiodic acid solution to generate rGO membranes. For the first time, the water permeability and Na2SO4 rejection of the rGO membrane reached the level of commercial polyamide-based nanofiltration membranes simultaneously. Hsin Hui Huang et al. [74] fabricated uniform rGO films by adjusting the hydrothermal reaction temperature and time through simple experiments.

References

  1. Hu, M.; Mi, B.X. Enabling graphene oxide nanosheets as water separation membranes. Environmental Science & Technology 2013, 47, 3715-3723.
  2. Liu, G.P.; Jin, W.Q.; Xu, N.P. Graphene-based membranes. Chemical Society Reviews 2015. 44, 5016-5030.
  3. Shao, F.F.; Xu, C.W.; Ji, W.B.; Dong, H.Z.; Sun, Q.; Yu, L.Y.; Dong, L.F. Layer-by-layer self-assembly TiO2 and graphene oxide on polyamide reverse osmosis membranes with improved membrane durability. Desalination 2017, 423, 21-29.
  4. Yi, M. Shen, Z.G. A review on mechanical exfoliation for the scalable production of graphene. Journal of Materials Chemistry A 2015, 3, 11700-11715.
  5. Ullah, S.; Liu, Y.; Hasan, M.; Zeng, W.W.; Shi, Q.T.; Yang, X.Q.; Fu, L.; Ta, H.Q.; Lian, X.Y.; Sun, J.Y.; Yang, R.Z.; Liu, L.J.; Rummeli, M.H. Direct synthesis of large-area Al-doped graphene by chemical vapor deposition: advancing the substitutionally doped graphene family. Nano Res 2022, 15, 1310-1318.
  6. Bae,S.; Kim, H.; Lee, Y.; Xu, X.F.; Park, J.S.; Zheng, Y.; Balakrishnan, J.; Lei, T.; Kim, H.R.; Song, Y.I.; Kim, Y.J.; Kim, K.S.; Ozyilmaz, B.; Ahn, J.H.; Hong, B.H.; Iijima, S. Roll-to-roll production of 30-inch graphene films for transparent electrodes, Nat Nanotechnol 2010, 5, 574-578.
  7. Kafiah, F.M.; Khan, Z.; Ibrahim, A.; Karnik, R.; Atieh, M.; Laoui, T. Monolayer graphene transfer onto polypropylene and polyvinylidenedifluoride microfiltration membranes for water desalination. Desalination 2016, 388, 29-37.
  8. Koenig, S.P.; Wang, L.D.; Pellegrino, J.; Bunch, J.S. Selective molecular sieving through porous graphene. Nat Nanotechnol 2012, 7, 728-732.
  9. Cohen-Tanugi, D.; McGovern, R.K.; Dave, S.H.; Lienhard, J.H.; Grossman, J.C. Quantifying the potential of ultra-permeable membranes for water desalination. Energy & Environmental Science 2014, 7, 1134-1141.
  10. Wang, H.; Zhang, D.S.; Yan, T.T.; Wen, X.R.; Shi, L.Y.; Zhang, J.P. Graphene prepared via a novel pyridine-thermal strategy for capacitive deionization. J Mater Chem 2012, 22, 23745-23748. 71. Nair, R.R.; Wu, H.A.; Jayaram, P.N.; Grigorieva, I.V.; Geim, A.K. Unimpeded permeation of water through helium-leak-tight graphene-based membranes. Science 2012, 335, 442-444.
  11. Sun, P.Z.; Zhu, M.; Wang, K.L.; Zhong, M.L.; Wei, J.Q.; Wu, D.H.; Xu, Z.P.; Zhu, H.W. Selective ion penetration of graphene oxide membranes. Acs Nano 2013, 7, 428-437.
  12. Wu, Z.S.; Ren, W.C.; Gao, L.B.; Liu, B.L.; Jiang, C.B.; Cheng, H.M. Synthesis of high-quality graphene with a pre-determined number of layers. Carbon 2009, 47, 493-499.
  13. Pei, J.X.; Zhang, X.T.; Huang, L.; Jiang, H.F.; Hu, X.J. Fabrication of reduced graphene oxide membranes for highly efficient water desalination. Rsc Advances 2016, 6, 101948-101952.

Reviewer 3 Report

This is an interesting manuscript. I would like to see this manuscript added to the research community, but just have a quick concern. All the papers were obtained by using keywords "Desalination and membrane and graphene" from Web of Science, and all the analysis was done by using these papers. However, the authors may miss lots of papers using only one database for the search. I understand that the authors explained why using Web of Science, but my question is that what is the reason that you did not consider using multiple database and other relevant keywords. Besides, if you obtained those papers using a certain keywords, it does not really make sense to me when you do "keywords analysis", because those papers were obtained by searching a certain keywords, and you will certainly get those keywords that you searched for in your "keywords analysis". Please explain how confident you are that your search and analysis are comprehensive and representative.

Author Response

Point 1: This is an interesting manuscript. I would like to see this manuscript added to the research community, but just have a quick concern. All the papers were obtained by using keywords "Desalination and membrane and graphene" from Web of Science, and all the analysis was done by using these papers. However, the authors may miss lots of papers using only one database for the search. I understand that the authors explained why using Web of Science, but my question is that what is the reason that you did not consider using multiple database and other relevant keywords.

Response 1: Thank you for the comment. Compared with other databases, Web of Science is a new generation of Internet-based academic resource integration platform of Clarivate Analytics. It is one of the oldest international databases and is considered a widely accepted and source authority for literature searches, citation analysis and journal indexing. The types of documents mainly include: journals, conference proceedings, patents and various resources in the academic website provided by the system. We use WOS as a database to discover trends in this direction. For example, some articles also use the database for data analysis.

[1] M.N. Naseer, K. Dutta, A.A. Zaidi, M. Asif, A. Alqahtany, N.A. Aldossary, R. Jamil, S.H. Alyami, J. Jaafar, Research Trends in the Use of Polyaniline Membrane for Water Treatment Applications: A Scientometric Analysis, Membranes-Basel, 12 (2022).

[2] D.C. Calvo, H.J. Luna, J.A. Arango, C.I. Torres, B.E. Rittmann, Determining global trends in syngas fermentation research through a bibliometric analysis, J Environ Manage, 307 (2022).

[3] M.N. Naseer, A.A. Zaidi, H. Khan, S. Kumar, M.T. Bin Owais, Y.A. Wahab, K. Dutta, J. Jaafar, M. Uzair, M.R. Johan, I.A. Badruddin, Desalination technology for energy-efficient and low-cost water production: A bibliometric analysis, Green Process Synth, 11 (2022) 306-315.

[4] A.K. Al-Buriahi, M.M. Al-shaibani, R.M.S.R. Mohamed, A.A. Al-Gheethi, A. Sharma, N. Ismail, Ciprofloxacin removal from non-clinical environment: A critical review of current methods and future trend prospects, J Water Process Eng, 47 (2022).

Of course, the reviewers' opinions are very pertinent. In our follow-up research, we will also consider analyzing the data of multiple databases to make the analysis results more comprehensive.

The search term we used was "Desalination and membrane and graphene", and the search terms were connected by Boolean logic operators. At the same time, the subject search is performed, which means that the search records including the title, abstract, author keywords, and words in Keywords Plus will be retrieved. This is a comprehensive search, not just a keyword checkout. Our retrieval formula contains the retrieval of the entire subject, which is representative.

For a clearer explanation, we have added the following statement to the manuscript:

The literature related to graphene-based membranes for water desalination was searched with the following text format: Topic= (Desalination) and (membrane) and (graphene) where topic search which implies looking for searched terms or schemes in the title, abstract, author keywords, and keywords plus.

Point 2: Besides, if you obtained those papers using a certain keywords, it does not really make sense to me when you do "keywords analysis", because those papers were obtained by searching a certain keywords, and you will certainly get those keywords that you searched for in your "keywords analysis". Please explain how confident you are that your search and analysis are comprehensive and representative.

Response 2: Thank you for the comment. For the retrieval strategy, we used the Boolean logic formula "Desalination and membrane and graphene", which was carried out by subject search, which refers to the title, abstract, author keywords, and words in Keywords Plus contained in the search records. In other words, article titles, abstracts, author keywords, and articles in Keywords Plus that contain this search strategy will all be retrieved. The keywords in all the articles obtained by the search not only contain the three keywords of "seawater desalination", "membrane" and "graphene" in the search formula. Besides the frequency of keywords, it is more important to analyze the co-occurrence network of keywords. The outcomes of this burst keywords are expected to help scholars in the desalination field find out the places where flowers are blooming the most.

For a clearer explanation, we have added the following statement to the manuscript:

The literature related to graphene-based membranes for water desalination was searched with the following text format: Topic= (Desalination) and (membrane) and (graphene) where topic search which implies looking for searched terms or schemes in the title, abstract, author keywords, and keywords plus.

Round 2

Reviewer 1 Report

The manuscript was revised to show its minimum quality for acceptance. 

One more comments about the cited publications. There are over 1739 articles about graphene materials and membrane for desalination, while only less than 100 articles are cited in this study. Is it good enough to cover the main topics or does the topic overexaggerate? 

Author Response

Point 1: The manuscript was revised to show its minimum quality for acceptance.

One more comments about the cited publications. There are over 1739 articles about graphene materials and membrane for desalination, while only less than 100 articles are cited in this study. Is it good enough to cover the main topics or does the topic overexaggerate?

Response 1: Thank you for the comment. The web of science database was used for the subject search. For the retrieval strategy, we used the Boolean logic formula "Desalination and membrane and graphene", which was carried out by subject search, which refers to the title, abstract, author keywords, and words in Keywords Plus contained in the search records. 1739 papers are all related to this topic. Subject searches were conducted through mainstream databases and we believe these documents are sufficient to cover the main subject. The manuscript is a mini review. For bibliometrics, we performed the analysis on all 1739 articles. According to the bibliometric results, the three most attractive directions were screened for review, so less than 100 articles were cited. We believe that these articles are representative in the mainstream directions. At the same time, we supplement the basic information of all literatures in the auxiliary materials of the paper for the convenience of readers.

We have supplemented the basic information of all literatures in the supporting materials of the thesis. For a clearer explanation, we have added the following statement to the manuscript:

The literature related to graphene-based membranes for water desalination was retrieved with the following text format: Topic= (Desalination) and (membrane) and (graphene) where topic search implies looking for searched terms or schemes in the title, abstract, author keywords, and keywords plus. Articles were based on the year from 2011 to 2022. The retrieved results can be found in the supporting information.